# SELF-ICL: Zero-Shot In-Context Learning with Self-Generated Demonstrations

**Wei-Lin Chen**[*]   **Cheng-Kuang Wu**[*]   **Yun-Nung Chen**   **Hsin-Hsi Chen**
National Taiwan University, Taiwan
{wlchen,ckwu}@nlg.csie.ntu.edu.tw
y.v.chen@ieee.org   hhchen@ntu.edu.tw

## Abstract

Large language models (LLMs) have exhibited striking in-context learning (ICL) ability to adapt to target tasks with a few input-output demonstrations. For better ICL, different methods are proposed to select representative demonstrations from existing training corpora. However, such settings are not aligned with real-world practices, as end-users usually query LMs without access to demonstration pools. In this work, we introduce SELF-ICL—a simple framework which bootstraps LMs' intrinsic capabilities to perform *zero-shot* ICL. Given a test input, SELF-ICL first prompts the model to generate pseudo-inputs. Next, the model predicts pseudo-labels for the pseudo-inputs via zero-shot prompting. Finally, we perform ICL for the test input with the pseudo-input-label pairs as demonstrations. Evaluation on 23 BIG-Bench Hard tasks shows SELF-ICL outperforms zero-shot baselines on both average accuracy and head-to-head comparison. Moreover, with zero-shot chain-of-thought, SELF-ICL achieves results comparable to using real demonstrations. Additionally, we conduct a range of analyses to validate SELF-ICL's effectiveness and provide insights for its behaviors under different settings.[1]

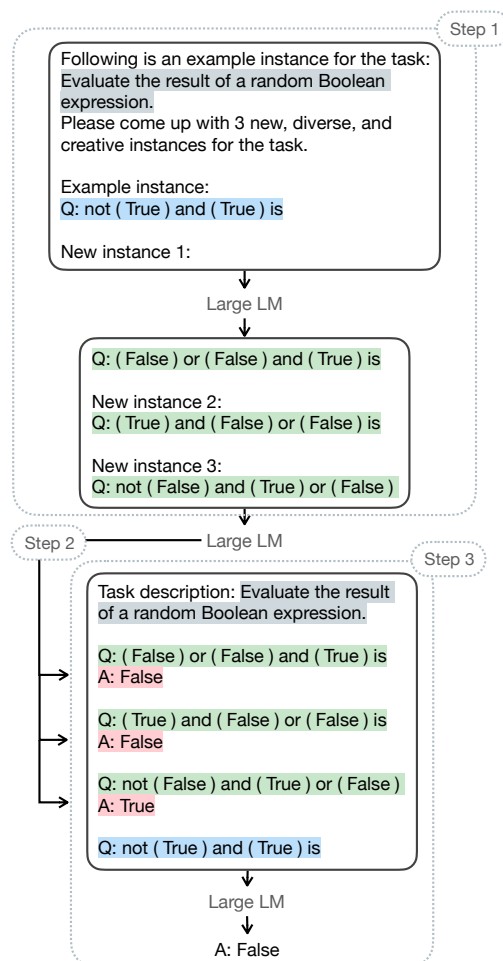

Figure 1: Our proposed SELF-ICL framework for zero-shot in-context learning. SELF-ICL involves three steps: (1) Given a query and a corresponding task description, the LLM is prompted to generate $k$ (e.g., $k$=3) pseudo-inputs. (2) Collect the pseudo-inputs and predict their pseudo-labels via zero-shot prompting. (3) Perform ICL with pseudo-demonstrations constructed from the generated pseudo-input-label pairs. The same LLM is used in all steps.

## 1 Introduction

Large language models (LMs) have shown striking ability to adapt to new tasks at test time by prompting with a few input-output exemplars, i.e., *demonstrations* (Brown et al., 2020; Wei et al., 2022; Chowdhery et al., 2022; Wei et al., 2023). This ability is refereed to as in-context learning (ICL; Brown et al., 2020). Towards better ICL performance, approaches for selecting representative demonstrations have been investigated extensively (Sorensen et al., 2022; Levy et al., 2022;

Zhang et al., 2022a; Gonen et al., 2022). Most techniques assume the access to large-scale external sources (e.g., training dataset or relevant text corpus) is available, from which demonstrations can

---

[*]Equal contribution.
[1]https://github.com/ntunlplab/Self-ICL

be selected with methods such as nearest neighbor search or other pre-defined, sophisticated similarity metrics (Liu et al., 2022; Rubin et al., 2022; Wu et al., 2022). However, in most real-world scenario, users query LLMs (e.g., through APIs or web interface) without the access to existing corpus for their target tasks. Also, spending additional effort to handcraft demonstrations may negatively affect their workflows.

Recently, a series of studies has been proposed to shed lights on the inner working of ICL (Xie et al., 2021; Reynolds and McDonell, 2021; Min et al., 2022b). The evidence suggests that instead of contributing explicit signals for learning new tasks, demonstrations mainly expose LLMs' intrinsic functionalities and guide models towards target domains (Razeghi et al., 2022; Lyu et al., 2022). Similar clues are also partly observed in chain-of-thought (CoT) and instruction-augmented ICL (Madaan and Yazdanbakhsh, 2022; Webson and Pavlick, 2022). These findings indicate, to some degree, LLMs carry underestimated zero-shot abilities and are already equipped to fulfill various target tasks.

Inspired by the above-mentioned literature, we propose SELF-ICL—a simple prompting framework for zero-shot in-context learning. SELF-ICL bootstraps LLM's intrinsic capabilities by self-generated demonstrations which inform the input and label space for performing ICL. Given a query, i.e., a test input, SELF-ICL's involves three steps:

1. The model is prompted to generate pseudo-inputs conditioned on the given query and the corresponding task description.

2. The model predicts pseudo-labels for pseudo-inputs via zero-shot prompting.

3. The pseudo-input-label pairs form pseudo-demonstrations, which are then prepended to the query and proceed with standard ICL.

All steps adopt the same frozen LLM. Without the requirement of candidate pool for demonstration selection, SELF-ICL bridges the gap for end-user's practical needs.

To evaluate SELF-ICL's effectiveness on challenging, unexpected tasks for which existing demonstrations are hard to come by, we perform evaluation on a set of 23 tasks from BIG-Bench Hard (BBH; Suzgun et al., 2022). Results show that SELF-ICL exhibits significant improvements on the

| Method | Inputs | Labels |
|---|---|---|
| AUTO-COT | from training set | *no need* |
| Z-ICL | from external corpus | *no need* |
| SG-ICL | *no need* | given |
| SELF-ICL (ours) | *no need* | *no need* |

Table 1: A comparison to prior attempts on zero-shot ICL. SELF-ICL does not require any real inputs or labels to construct demonstrations. See Section 6 for detailed discussions.

all-task-average accuracy and in head-to-head comparisons. For instance, the results are 18-0-5 (win-tie-lose) for SELF-ICL versus standard zero-shot on the 23 tasks. Furthermore, with *zero-shot Chain-of-Thought* (Kojima et al., 2022), SELF-ICL reaches performance on par with using few-shot demonstrations sampled from real data instances.

In addition, we perform an array of analyses to validate SELF-ICL's effectiveness under different settings. We investigate various approaches for generated pseudo-inputs, the effect of number of shots, and the impact of random pseudo-labels, providing better insights for SELF-ICL's behaviours. To the best of our knowledge, we present the first attempt for *true* zero-shot ICL that does not require any external data from real distribution or pre-defined label sets (See Table 1).

## 2 SELF-ICL

This section details the design of SELF-ICL for constructing pseudo-inputs and pseudo-labels to form ideal pseudo-demonstrations.

### 2.1 Pseudo-Input Construction (Step 1)

Generating pseudo-inputs can be easily achieved by zero-shot prompting LLMs with the simple prompt as shown in Figure 1 (Step 1). The given query $q$ (from real distribution) provides an outline of ground-truth inputs, and the corresponding task description $T$ guides the model to generate relevant information associated with the task domain. From $q$ and $T$, model infers the underlying format and creates a new query (i.e., pseudo-input). By specifying a number $k$ (number of shots) in the instruction, this process can generate multiple pseudo-inputs with one inference pass.

### 2.2 Pseudo-Label Construction (Step 2)

After obtaining the pseudo-inputs, we then predict their labels (the pseudo-labels for constructing pseudo-demonstrations) via zero-shot prompting

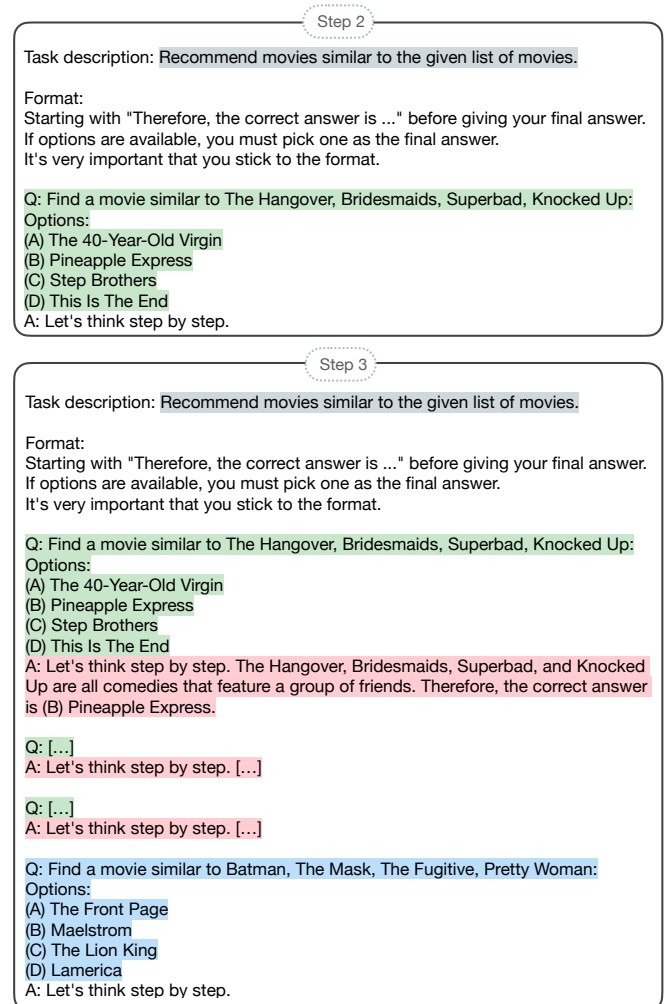

Figure 2: Example prompts of SELF-ICL Steps 2 and 3 for the CoT prompting setting (*movie recommendation*).

the same LLM. Specifically, we employ two zero-shot methods: *Direct prompting* and *CoT prompting*, described as follows.

**Direct prompting**  In the direct prompting setup, we construct pseudo-labels via standard zero-shot prompting schema. Namely, we prompt the LLM with only the task description and the generated pseudo-input for a direct answer prediction (See Figure 8 for an example prompt). We predict pseudo-labels one by one, i.e., for $k$-shot demonstration, $k$ inference passes are required for the $k$ pseudo-inputs.

**CoT prompting**  For the CoT prompting setup, SELF-ICL generates pseudo-labels by *zero-shot CoT* (Kojima et al., 2022). Specifically, we prompt the LLM with the task description, the current test input, and a trigger phrase, "*Let's think step by step.*" for performing CoT reasoning. The trigger phrase is appended at the very end of the prompt,

guiding the model to generate its intermediate reasoning steps which lead to a more accurate final answer. We then take the trigger phrase and the generated reasoning chain containing the answer for pseudo-inputs as the pseudo-labels for constructing pseudo-demonstrations (See Figure 2 for an example prompt).

## 2.3 Prediction (Step 3)

Here in Step 3, we construct pseudo-demonstrations, i.e., pseudo-shots, by the pseudo-inputs paired with their corresponding pseudo-labels from previous steps, and predict the final answer for the test input by the typical few-shot ICL workflow. Namely, the pseudo-shots (with instructions) are prepended to the test input as the context for prompting the LLM. For CoT prompting, only the final answers are evaluated. For the example prompts on Step 3, see Figure 8 and 2. Note that both direct prompting and CoT

| BBH Task | Direct Prompting | | | CoT Prompting | | | 3-shot |
|---|---|---|---|---|---|---|---|
| | ZS-Direct | Self-ICL | $\delta$ | ZS-CoT | Self-ICL | $\delta$ | |
| Boolean Expressions | 84.00 | **87.20** | 3.20 | 85.60 | **85.60** | 0.00 | 89.60 |
| Causal Judgement | 55.61 | **61.50** | 5.88 | 58.29 | **58.82** | 0.53 | 62.03 |
| Date Understanding | 54.00 | **56.80** | 2.80 | 66.80 | **69.20** | 2.40 | 58.80 |
| Disambiguation QA | 64.00 | **68.00** | 4.00 | 64.80 | **67.60** | 2.80 | 68.80 |
| Formal Fallacies | 56.00 | 55.20 | -0.80 | 55.20 | **56.80** | 1.60 | 58.80 |
| Geometric Shapes | 34.40 | **36.00** | 1.60 | 29.60 | **33.60** | 4.00 | 36.80 |
| Hyperbaton | 56.80 | **59.20** | 2.40 | 53.60 | **53.60** | 0.00 | 57.60 |
| Logical Deduction (five objects) | 41.20 | 40.40 | -0.80 | 37.60 | **40.80** | 3.20 | 45.60 |
| Logical Deduction (seven objects) | 43.60 | 40.00 | -3.60 | 39.60 | 36.00 | -3.60 | 40.00 |
| Logical Deduction (three objects) | 56.00 | **65.60** | 9.60 | 58.80 | **64.80** | 6.00 | 64.40 |
| Movie Recommendation | 63.60 | **75.20** | 11.60 | 64.80 | **70.40** | 5.60 | 77.20 |
| Navigate | 49.20 | **68.00** | 18.80 | 53.60 | **57.60** | 4.00 | 52.80 |
| Penguins in a Table | 58.90 | **63.70** | 4.79 | 60.96 | **64.38** | 3.42 | 62.33 |
| Reasoning about Colored Objects | 59.60 | **61.20** | 1.60 | 68.00 | 67.60 | -0.40 | 65.20 |
| Ruin Names | 54.40 | **66.40** | 12.00 | 48.80 | **56.40** | 7.60 | 84.00 |
| Salient Translation Error Detection | 51.20 | **57.20** | 6.00 | 50.80 | **54.00** | 3.20 | 65.20 |
| Snarks | 54.49 | **62.92** | 8.43 | 37.08 | **55.06** | 17.98 | 65.73 |
| Sports Understanding | 67.60 | 65.60 | -2.00 | 71.20 | 69.20 | -2.00 | 71.20 |
| Temporal Sequences | 57.60 | 35.60 | -22.00 | 64.80 | 52.00 | -12.80 | 39.20 |
| Tracking Shuffled Objects (five objs) | 18.00 | **19.60** | 1.60 | 25.20 | **29.60** | 4.40 | 16.40 |
| Tracking Shuffled Objects (seven objs) | 14.40 | **18.40** | 4.00 | 31.60 | 27.20 | -4.40 | 15.20 |
| Tracking Shuffled Objects (three objs) | 26.40 | **28.00** | 1.60 | 36.00 | **46.00** | 10.00 | 30.40 |
| Web of Lies | 53.20 | **57.20** | 4.00 | 61.20 | **65.60** | 4.40 | 56.40 |
| All Tasks (avg) | 50.81 | **53.93**[†] | 3.12 | 53.22 | **55.54**[†] | 2.32 | 55.49 |

Table 2: The main results of our proposed SELF-ICL evaluated on BBH. The *3-shot* results are standard few-shot prompting with real data as demonstrations. SELF-ICL exhibits consistent trends outperforming both direct and CoT prompting baselines. We adopt one-sided McNemar's test (McNemar, 1947) to test the statistical significance of Self-ICL's performance gain over baselines, where † denotes $p$ value < 0.05.

prompting methods shared the same Step 1 prompt (Figure 7).

# 3 Experiments

To evaluate the effectiveness of our proposed method, we conduct a set of extensive experiments for better comparison and analysis. We describe the experimental settings and discuss the results in detail.

## 3.1 Configurations

**Language models** We use InstructGPT (`text-davinci-003`; Ouyang et al., 2022) for all the experiments presented in Section 4.1 and 5. We also conduct additional experiments to validate the generalizability of SELF-ICL, using `text-bison-001`[2] from the PaLM-2 model family (Anil et al., 2023) and `gpt-3.5-turbo-instruct` from the GPT-3.5 model family.[3] The results are presented in Section 4.2.

[2]https://developers.generativeai.google/models/language
[3]https://platform.openai.com/docs/models/gpt-3-5

**Implementation details** For all LMs' hyperparameters, we set the temperature to 0 and the the maximum number of tokens as 1024. Other arguments are kept as their default values. Regarding the number of pseudo-demonstration shots $k$, we choose $k=3$ for our main experiments.

**Dataset** We adopt the BIG-Bench Hard (BBH) benchmark for our evaluation. BBH consists of a suite of tasks from the BIG-Bench benchmark (Srivastava et al., 2022), which existing LMs have difficulty reaching the average human-rater performance and are considered beyond current models' capabilities. BBH contains a total of 27 tasks, from which we select 23 tasks that are multiple-choice tasks as our evaluation testbed for SELF-ICL. Each BBH tasks has around $150 \sim 250$ examples, and the total number of instances is 5,511.

## 3.2 Baselines

**ZS-Direct** The baseline of direct prompting is the typical zero-shot prompting setup, denoted as ZS-Direct. Concretely, the LLM is prompted with the task description and the current test input for a direct answer prediction.

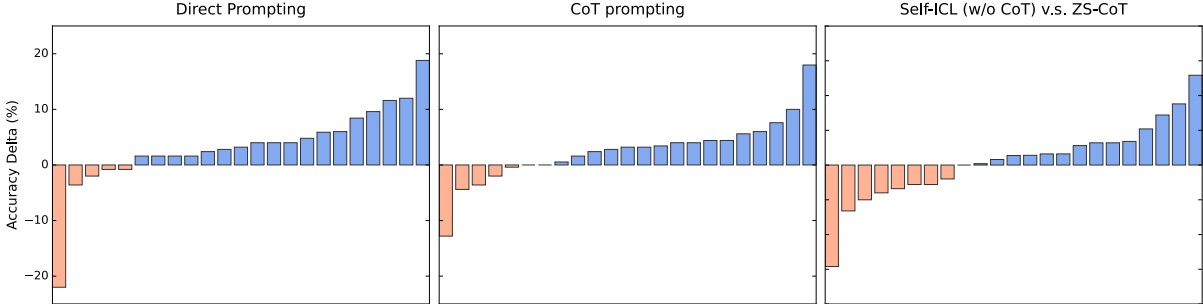

Figure 3: The head-to-head comparison on the 23 tasks from BBH. The accuracy delta indicates the accuracy difference between SELF-ICL and the baseline method (blue/orange indicates our method wins/loses). The results are 18-0-5 (win-tie-lose) for the direct prompting setting; 16-2-5 for the CoT prompting setting; and 14-1-8 for SELF-ICL *without* CoT (i.e., direct) versus ZS-CoT.

**ZS-CoT** For CoT prompting, the baseline is the zero-shot CoT prompting proposed by Kojima et al. (2022), which is one of the state-of-the-art method for solving reasoning-heavy task in zero-shot. We denoted it as ZS-CoT. Specifically, the LLM is prompted by the task description, the current test input, and a reasoning trigger phrase "*Let's think step by step.*" same as in SELF-ICL.

## 4 Results

### 4.1 Main Results

We present our main experimental results in Table 2. On the *all tasks average* performance, SELF-ICL surpasses baselines in both the direct and CoT prompting settings. We also observe SELF-ICL with direct prompting is comparable (slightly better) with ZS-CoT prompting. Furthermore, SELF-ICL with CoT prompting reaches performance on par with few-shot prompting which uses real demonstrations (the *3-shot* column).

We illustrate head-to-head comparisons on the 23 tasks in Figure 3. The results of direct prompting are 18-0-5 (win-tie-lose) for SELF-ICL versus the ZS-Direct baseline; for the CoT prompting setting, the results are 16-2-5 for SELF-ICL versus the ZS-CoT baseline. Interestingly, the results are 14-1-8 for SELF-ICL *without* CoT (SELF-ICL with direct prompting) versus ZS-CoT, and comparable or better performance is exhibited on *all tasks average* as well. This highly competitive result demonstrated by SELF-ICL with direct prompting sheds light on an alternative to elicit LMs' reasoning ability in zero-shot, without generating potentially biased or misleading reasoning chains (Turpin et al., 2023).

### 4.2 Generalizability

To assess whether our proposed SELF-ICL framework is able to generalize to other models, we perform experiments on two popular LLMs, GPT-3.5 and PaLM-2, beside InstructGPT. We compare their results under the direct prompting setting. The results are present in Table 3. As observed, SELF-ICL demonstrates stronger overall performance over the direct prompting baseline on both PaLM-2 and GPT-3.5. Moreover, although PaLM-2 exhibits relatively poor scores comparing to InstructGPT and GPT-3.5, it can still improve upon itself with our proposed SELF-ICL. Interestingly, we find GPT-3.5 has a slightly inferior performance comparing to InstructGPT. We hypothesize this is because GPT-3.5 has a lower controllability, thus, it is more prone to generate unintended content. For instance, the model might not follow the formatting instructions presented in the prompts (see Figure 8). In addition, the generated pseudo-inputs are more likely to be invalid and could not accurately represent the underlying tasks. In sum, the results still suggests SELF-ICL is generalizable for different models.

## 5 Analysis

In this section, we first illustrate the concept of *copying effect*, and discuss its implication for SELF-ICL. Next, we investigate SELF-ICL's behaviors under different settings, including different approaches for generated pseudo-inputs, performance with varied number of shots, and the effect of randomly assigning pseudo-labels. Following analyses all focus on the setting of direct prompting.

| BBH Task | text-bison-001 | | | gpt-3.5-turbo-instruct | | |
|---|---|---|---|---|---|---|
| | ZS-Direct | Self-ICL | $\delta$ | ZS-Direct | Self-ICL | $\delta$ |
| Boolean Expressions | 60.16 | 58.94 | -1.22 | 84.80 | **88.40** | 3.60 |
| Causal Judgement | 47.37 | **48.54** | 1.17 | 42.25 | 12.30 | -29.95 |
| Date Understanding | 42.40 | 41.20 | -1.20 | 59.20 | 57.60 | -1.60 |
| Disambiguation QA | 33.33 | **33.82** | 0.49 | 60.00 | **63.20** | 3.20 |
| Formal Fallacies | 57.20 | 56.00 | -1.20 | 52.00 | 50.40 | -1.60 |
| Geometric Shapes | 15.20 | **19.20** | 4.00 | 34.00 | **36.40** | 2.40 |
| Hyperbaton | 57.43 | **68.27** | 10.84 | 82.40 | **82.80** | 0.40 |
| Logical Deduction (five objects) | 18.34 | **23.58** | 5.24 | 42.00 | 38.40 | -3.60 |
| Logical Deduction (seven objects) | 10.88 | **13.99** | 3.11 | 41.60 | 34.80 | -6.80 |
| Logical Deduction (three objects) | 37.89 | 37.00 | -0.88 | 56.00 | **59.20** | 3.20 |
| Movie Recommendation | 26.23 | **26.23** | 0.00 | 74.80 | **76.00** | 1.20 |
| Navigate | 58.71 | 57.21 | -1.49 | 42.80 | **64.80** | 22.00 |
| Penguins in a Table | 42.76 | **42.76** | 0.00 | 51.37 | **55.48** | 4.11 |
| Reasoning about Colored Objects | 62.40 | **70.80** | 8.40 | 54.80 | **56.40** | 1.60 |
| Ruin Names | 30.81 | 26.16 | -4.65 | 70.80 | 64.80 | -6.00 |
| Salient Translation Error Detection | 22.13 | **22.54** | 0.41 | 41.60 | **51.20** | 9.60 |
| Snarks | 54.86 | 50.86 | -4.00 | 63.48 | 60.67 | -2.81 |
| Sports Understanding | 46.45 | 45.90 | -0.55 | 62.00 | 50.00 | -12.00 |
| Temporal Sequences | 28.51 | **30.58** | 2.07 | 20.80 | **32.80** | 12.00 |
| Tracking Shuffled Objects (five objects) | 14.52 | **17.74** | 3.23 | 18.00 | 16.40 | -1.60 |
| Tracking Shuffled Objects (seven objects) | 20.00 | 19.60 | -0.40 | 17.60 | 12.40 | -5.20 |
| Tracking Shuffled Objects (three objects) | 29.32 | **32.93** | 3.61 | 32.40 | **36.80** | 4.40 |
| Web of Lies | 57.20 | 50.40 | -6.80 | 15.20 | **38.40** | 23.20 |
| All Tasks (avg) | 37.78 | **38.83**$^\dagger$ | 1.05 | 48.52 | **49.72**$^\dagger$ | 1.20 |

Table 3: The results of SELF-ICL using `text-bison-001` and `gpt-3.5-turbo-instruct` evaluated on BBH. Overall, SELF-ICL exhibits consistent trends outperforming direct prompting. This suggests SELF-ICL is generalizable for different models. We adopt one-sided McNemar's test (McNemar, 1947) to test the statistical significance of Self-ICL's performance gain over baselines, where $\dagger$ denotes $p$ value < 0.05.

## 5.1 Preliminary

Given a set of $k$-shot demonstrations denoted as $\{(x_1, y_1), ..., (x_k, y_k)\}$, where $x_i$ is the input text and $y_i$ is the label. As suggested by Min et al. (2022b), four aspects are considered for the construction of demonstrations, namely: (1) The *input-label mapping*: whether $x_i$ is paired with a correct $y_i$. (2) The *input space*: the underlying distribution behind $x_1, ..., x_k$. (3) The *label space*: the possible label set inferred from $y_1, ..., y_k$.[4] (4) The *pairing format*: the format representing the $x_i$-$y_i$ pair. Min et al. (2022b) inspect the role of demonstrations along these four aspects, and present a surprising finding—the input-label mapping is *not* a necessary criteria for successful ICL. Empirically, they find randomly swapping the ground-truth label of demonstrations barely degrades end-task performance. On the contrary, the other three aspects all demonstrate great impacts. With these four aspects in mind, we now analyze the construction of pseudo-demonstrations for SELF-ICL.

## 5.2 The Entanglement of *Input Space* and *Input-Label Mapping*

Among the four aspects, the *label space* is usually specified in the input (e.g., options presented for multiple-choice) or described in the task description. For example, the label space {*"True"*, *"False"*} of the *boolean expressions* task can be easily inferred from its description *"Evaluate the result of a random Boolean expression."*; the *pairing format* is the least concern as pseudo-demonstrations are well formatted as "Q: *input text*, A: *label text*".

The potentially problematic aspects are the *input space* and *input-label mapping*. Naturally, one may think input-label mapping is not an issue as described in Section 5.1—the input does not need to be paired with the correct label. However, this intriguing discovery by Min et al. (2022b) is established under the setting of standard ICL, where the inputs are randomly sampled from the training set.

As the pseudo-inputs created by SELF-ICL is based on only one reference, i.e., the given test input, the generated pseudo-inputs are likely to be of great semantic similarity with that test input, and fail to capture the correct input space distri-

---

[4]Input with instruction-like descriptions could also inform model of the label space.

bution. In such case, Min et al. (2022b)'s finding does not hold since it has been shown that models tend to *copy* the labels paired with inputs that are very similar to the test input, known as the *copying effect* (Lyu et al., 2022). With no guarantee for the correctness of SELF-ICL's pseudo-labels, the copying effect would potentially hurt the ICL performance.

## 5.3 Different Approaches for Generating Pseudo-Inputs

To mitigate the possible impact of copying effect, increasing the pseudo-inputs' diversity is essential. Typically, this can be resolved by sampling demonstration inputs from different clusters of training set inputs (Zhang et al., 2022b). However, no real data is available in our SELF-ICL framework. To gain a better understanding of SELF-ICL's pseudo-input generation and the potential copying effect, we study three different approaches for constructing pseudo-inputs: (1) Batch inference, (2) Prompting *with* diversity hints, and (3) Prompt *without* diversity hints.

**Batch inference**   In batch inference, we assume an access to multiple test inputs in Step 1. Specifically, the number of example instances in the prompt equals the number of given test input, i.e., the batch size. The LM then generates the same number of pseudo-inputs as in the original streaming inference where we prompt one test input at a time. The prompting template is provided in Figure 9. In batch inference setup, all test inputs share the same pseudo-inputs, thus the same pseudo-demonstrations in Step 3.

**Prompting *with* diversity hints**   Prompting with diversity hints is the method we adopt in our main experiments. As shown in Figure 1 (Step 1), the model is explicitly instructed to provide "*new*", "*diverse*", and "*creative*" pseudo-input instances.

**Prompting *without* diversity hints**   For prompting without diversity hints, we simply remove the key words "*new*", "*diverse*", and "*creative*"in the instruction, and keep all other settings unchanged.

Our analysis results are shown in Figure 4. We compute the cosine similarity between the pseudo-inputs generated by the aforementioned approaches and the test input. For each method, the reported value is averaged across three pseudo-inputs (3-shots) and all BBH tasks. We also report the result

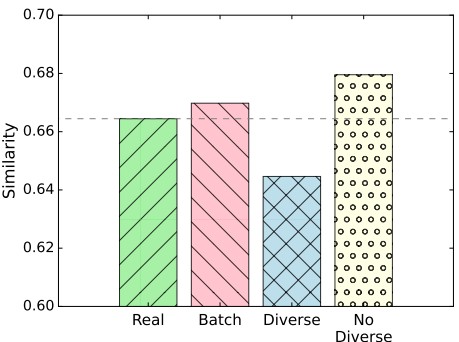

Figure 4: Semantic similarity between the pseudo-inputs generated by different Step 1 approaches and the test input. The similarity value is averaged across three shots and all tasks.

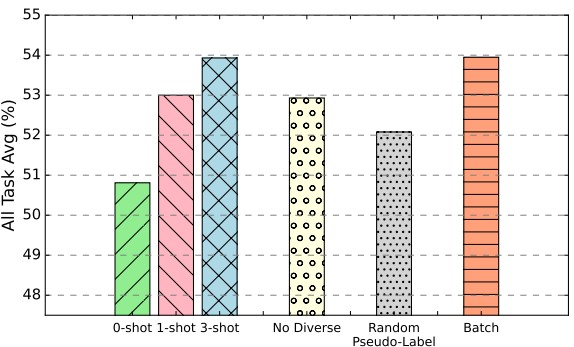

Figure 5: The all-task-average performance of using pseudo-inputs generated by different Step 1 approaches, different number of shots, and random pseudo-labels in Step 2.

of using real inputs from BBH dataset for establishing a similarity baseline. We encode all inputs by sentence transformers (Reimers and Gurevych, 2019) following Liu et al. (2022); Zhang et al. (2022b).[5]

As observed, batch inference produces pseudo-inputs that is most similar to using real inputs. This is somewhat intuitive as batch inference has access to more example instances. Interestingly, looking at results in Figure 5 we find using pseudo-inputs generated by prompting with diversity hints (the 3-shot bar) and batch inference achieve essentially the same final accuracy, although it exhibits the lower similarity. This may suggest over diversifying demo-inputs have little impact on empirical performance. For prompting without diversity hints, it demonstrates the highest similarity to test input and lower final accuracy, which could be explained by copying effect.

---

[5]We adopt the *all-MiniLM-L6-v2* model.

## 5.4 Effect of Different Number of Shots

Here we investigate SELF-ICL's performance under different number of pseudo-demonstration shots. The results are presented in Figure 5. The 3-shot setting is our adopted method in SELF-ICL's main experiment, and 1-shot setting used a randomly sampled shot from the 3 shots (here a shot refers to a pseudo-demonstration). The 0-shot setting is the ZS-Direct baseline. As observed, the 3-shot is the top performing setup. Note that although inferior to 3-shot, 1-shot still exhibits a notable gain over 0-shot, indicating the empirical effectiveness of SELF-ICL.

## 5.5 Effect of Random Pseudo-Labels

To verify the quality of our pseudo-labels, we replace the pseudo-labels obtained in step 2 by randomly assigned labels, and construct pseudo-demonstration with such random pseudo-labels to predict the test inputs' answers. As shown in Figure 5, the performance with random pseudo-labels is inferior to 3-shot, 1-shot, and the no diverse setup, but still benefit the performance comparing to no demonstration at all (0-shot).

Although the performance drop using random labels may indicate the possibility that some instances encounter the copying effect, we hypothesize the LLMs' abilities to overwrite the semantic prior when demonstrations have contradicting labels is another big factor (Wei et al., 2023). That is, LLMs would recognize the demonstration labels as the correct answers, and make predictions accordingly. Moreover, this phenomenon is further extrapolated when using LMs with instruction tuning. Exploring the underlying relationship between the copying effect and Wei et al. (2023)'s findings are left as future works.

## 5.6 A Deeper Look of SELF-ICL's Pseudo-Inputs

To increase the diversity of the generated pseudo-inputs and mitigate the risk of facing the copying effect, we apply a simple and straightforward method: prompting LLMs to be diverse with key words "*new*", "*diverse*", and "*creative*". To provide a more fine-grained analysis for individual tasks, following we attempt to quantitatively verify whether our generated pseudo-inputs are diverse enough in comparison with the real inputs randomly sampled from the training data, by measuring the *similarity gap* of the query-input distance

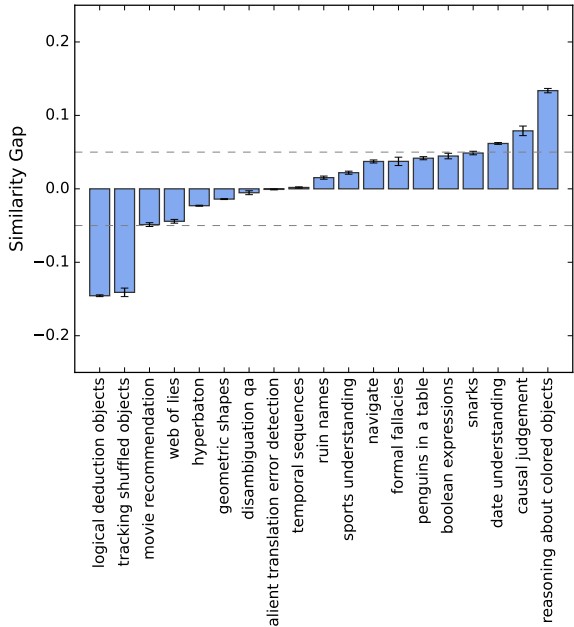

Figure 6: The similarity gap of the query-inputs distance between pseudo- and real-inputs. Most tasks fall into a samll $\pm 5\%$ range (the dotted lines), indicating the pseudo-inputs are close to the real-inputs and are likely robust against the copying effect.

between pseudo- and real-inputs.

Given a query $q$, i.e., test input, a set of $k$ randomly selected inputs $\{(x_1, y_1), ..., (x_k, y_k)\}$ from the training set, and a set of $k$ pseudo-inputs $\{(\hat{x}_1, \hat{y}_1), ..., (\hat{x}_k, \hat{y}_k)\}$ generated by SELF-ICL conditioned on $q$. We first define the query-input distance $d(\cdot)$ between using pseudo-input and real-input as

$$d(q) = \frac{1}{k}\sum_{i=1}^{k}\text{sim}(q, \hat{x}_i) - \frac{1}{k}\sum_{i=1}^{k}\text{sim}(q, x_i) \quad (1)$$

where the query and input are encoded by the same sentence transformers model used in Section 5.3. Next, we compute the similarity gap $\mathcal{G}(\cdot)$ as

$$\mathcal{G}(Q) = \frac{1}{n}\sum_{i=1}^{n}d(q_i) \quad (2)$$

where $Q$ is a set of $n$ queries $\{q_1, ..., q_n\}$ for a task.

The similarity gaps for 23 tasks from BBH are presented in Figure 6. The results are averaged across five different random seeds (for training data sampling) and we provide standard deviation error bars. The larger the gap indicates the more closer the queries are with the pseudo-inputs than with real-inputs sampled from training set, and the more likely to suffer from the copying effect. As

observed, most of the tasks fall inside the $\pm 5\%$ similarity range (dotted lines), suggesting our designed prompt is able to encourage the generation of diverse pseudo-inputs, and sufficiently resemble inputs sampled from real distributions to mitigating the potential risk of the copying effect. We also observe the three tasks with substantially higher or lower similarity gap require heavy, multi-step reasoning to solve. Thus the initial difficulties of understanding those task could explain model's failure to capture suitable input spaces.

# 6    Related Work

**Understanding ICL**   With the popularization of various LLMs, ICL has emerged as a new paradigm for the field of natural language processing (NLP). However, the mechanisms behind ICL's superior ability are still an open question in the research communities. To develop a deeper understanding of ICL, Chan et al. (2022) investigate the training data distribution of LLMs, and find specific distributional properties and the transformer-based architecture (Vaswani et al., 2017) could drive the ICL behaviors. Recent studies also provide explanations viewing LMs as meta-optimizers with meta-gradients applied in the forward passes, and show evidence of resemblances between ICL and the explicit fine-tuning process (Dai et al., 2022; von Oswald et al., 2022; Akyürek et al., 2022).

**Towards Zero-Shot ICL**   To achieve better empirical performance with ICL, approaches for designing ideal prompts and demonstrations have been vastly investigated (Min et al., 2022a; Su et al., 2022; Zhou et al., 2022; Lu et al., 2022a; Fu et al., 2022; Lu et al., 2022b).

Recent work from Zhang et al. (2022b) addressed the need of human-annotated few-shot CoT by utilizing zero-shot CoT to construct demonstrations. Their method differs from ours as they require an existing training set from which shots are sampled as inputs to zero-shot CoT. Lyu et al. (2022) attempt to exclude the need of pre-given demonstration candidate set by selecting semantically relevant sentences from an raw text corpus (which is not from the task datasets) as pseudo-inputs. And pair the selected pseudo-inputs with randomly assigned labels as demonstrations for ICL. Though more similar to our setting, they still need an access to external sources for constructing pseudo-inputs. Moreover, they are limited to classification tasks where a fixed set of labels is

shared among all inputs. On the contrary, SELF-ICL generates different input-dependent options for the multiple-choice tasks, and can easily extend to other generation tasks.

The most similar work to ours is by Kim et al. (2022), where they explore the possibility of generating pseudo-inputs by the LLM itself, without any external data source. However, their framework requires assess to the label set. They generate the pseudo-input by conditioning the LM on a label given in the prompt. Such a design dose not align with practical usage as it greatly restricts the scenario to fixed classification tasks. As a result, their evaluation is limited to only text classifications (sentiment classification and natural language inference), which are relatively simple and well-studied comparing to BBH in our evaluation.

# 7    Conclusions

In this work, we introduce SELF-ICL—a simple yet effective framework for zero-shot in-context learning, where only a test input and its task description are required. SELF-ICL consists of three steps: (1) Construction of pseudo-inputs, (2) Construction of pseudo-labels, (3) ICL with pseudo-demonstrations, i.e., pseudo-input-label pairs. Evaluations on BBH show SELF-ICL outperforms zero-shot (CoT) baselines on head-to-head and all-task average accuracy. Additionally, we conduct extensive analyses to provide a better insight of SELF-ICL. To the best of our knowledge, we present the first *true* zero-shot approach for ICL, and demonstrate the potential of bootstrapping LMs' inner capabilities to improve zero-shot performance.

## Limitations

**Reliance on instruction-following models**   To follow instructions, understand unseen target tasks and generate pseudo-inputs and pseudo-labels via zero-shot prompting, a key driver of our SELF-ICL framework is the powerful instruction-following LM. If the model is not equipped with such zero-shot generalization capability, the results of SELF-ICL would be inferior.

**Better diversify approaches**   To mitigate potential risks of suffering from the copying effect, we simply construct heuristic prompts to *tell* the LM to generate diverse pseudo-inputs. Due to the limited budget, we do not perform comprehensive prompt searching or experiment with temper-

ature adjustments. In the future, others should explore methods along the line of one- or few-shot data augmentation for constructing optimal pseudo-demonstrations.

## Acknowledgements

We thank the reviewers for their insightful comments. This work was financially supported by the National Science and Technology Council (NSTC) in Taiwan, under Grants 111-2222-E-002-013-MY3, 112-2223-E-002-012-MY5, 110-2221-E-002-128-MY3 and 111-2634-F-002-023, and Ministry of Education (MOE) in Taiwan, under grants NTU-112L900901.

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

# A Appendix

## A.1 Details of Experimental Cost

| Direct Prompting | | |
|---|---|---|
| | 3-shot | 118.35 |
| | 1-shot | 27.29 |
| SELF-ICL | prompting without diversity hints | 135.58 |
| | random pseudo-labels | 51.27 |
| | batch inference | 63.15 |
| Standard | 3-shot | 49.98 |
| | 0-shot | 15.27 |
| CoT Prompting | | |
| SELF-ICL | 3-shot | 203.10 |
| Standard | 0-shot | 28.71 |
| **Sum** | | **692.70** |

Table 4: The cost (in US dollar) of experiments using InstructGPT (Section 4.1 and 5), estimated based on number of tokens.

## A.2 Example Prompts

Figure 7: An example prompt of SELF-ICL Step 1 (The *movie recommendation* task). Note that both the direct prompting and CoT prompting settings shared the exact same Step 1 prompt.

Figure 8: Example prompts of SELF-ICL Steps 2 and 3 for the the direct prompting setting (The *movie recommendation* task).

Figure 9: The prompt template of batch inference (Step 1).