# OpenReview forum: "Self-ICL: Zero-Shot In-Context Learning with Self-Generated Demonstrations"
_EMNLP/2023/Conference — EMNLP 2023 Main_

### Official Review · Reviewer_aCDP · 2023-08-01

**Soundness:** 4

**Excitement:**

3: Ambivalent: It has merits (e.g., it reports state-of-the-art results, the idea is nice), but there are key weaknesses (e.g., it describes incremental work), and it can significantly benefit from another round of revision. However, I won't object to accepting it if my co-reviewers champion it.

**Paper Topic And Main Contributions:**

This paper proposes Self-ICL, a method for in-context learning in a zero-shot setup. Specifically, the model is first prompted to generate some pseudo-inputs and the model is then asked to respond to these pseudo-inputs to obtain a sequence of pseudo-demonstrations. The pseudo-demonstrations are then concatenated to the test queries in a manner similar to few-shot in-context learning to perform inference. The authors show effectiveness of their method on the BIG-Bench Hard problems with popular GPT models.

**Reasons To Accept:**

- The methodology is sensible, and I also see the value of this problem setting of extending in-context learning to a strictly zero-shot setup; previous works, like AutoCoT, are *transductive* zero-shot as unlabeled data are typically required -- I do have some concerns and questions regarding this, please see *Reasons to Reject* below.

- Experiments are largely thorough, and the choice of the datasets (the entire BIG-Bench Hard suite of 25 tasks) and models is convincing. The outperformance over the baselines seems significant, which is also supported by statistical tests.


**Reasons To Reject:**

- Related to the first point I mentioned in *Reasons to Accept*, my concern is as follows: while I see the academic value of the extension to ICL to a strictly zero-shot setup, I wonder whether the authors could clarify the practical significance?
  - The proposed method requires the user to prompt the LLM to generate pseudo-inputs and answers to *each* of the test questions, and the user is then required to concatenate the pseudo-demos to the test queries. In contrast, competing methods like AutoCoT or [1] generate a set of pseudo-demos *per task* but requires the user to provide an unlabelled dataset (which is cheap). On balance, it seems there is still a significant burden on the users, and since the users now need to prompt for model-generated pseudo-inputs *and* the final predictions, the users need to query the model many more times, so I'm not sure to what extent the claim of "bridging user's practical need" is satisfied.

  - Moreover, AutoCoT paper also has a section on the streaming setup where the unlabeled queries arrive in an online fashion; this is a practical enough setup, as we can always collect unlabeled user queries when the model is serving requests. I am requesting the authors to clarify better in what scenarios are their purely zero-shot setup particularly advantageous compared to these previous works?

- Related to what I mentioned above, given that we need to prompt the LLM more times per test query, the method is also more expensive. On the other hand, the baselines, such as zero-shot-CoT and zero-shot-direct, do not take advantage of the higher budget at all. One simple but effective way to improve the performance of the baselines is through self-consistency [2], especially for the CoT tasks. I wonder whether the authors could 1) state how much more expensive their method is compared to zs-cot and zs-direct, and 2) if the baselines are allowed to use the same amount of compute via self-consistency, is their method still better than the baselines? I think some evidence of this could alleviate the concern that the improvement is simply due to more computation used and can be easily bridged with a simpler alternative.

- Not a reason to reject, but simply a request for authors to comment: [1] is considered to be a concurrent work, but it considers a very related problem of using ICL in (transductive) zero-shot (similar to AutoCoT). They focus on a different angle in retrieving the best zero-shot demonstrations using a certainty estimate from self-consistency. For example, will the proposed method by the authors benefit from a similar selection procedure? I'd much appreciate if the authors could include discussions of these more recent related works in a revised version of their manuscript.

### References

[1]  Xingchen Wan, Ruoxi Sun, Hanjun Dai, Sercan Arik, and Tomas Pfister. 2023. Better Zero-Shot Reasoning with Self-Adaptive Prompting. In Findings of the Association for Computational Linguistics: ACL 2023, pages 3493–3514, Toronto, Canada. Association for Computational Linguistics.

[2] Wang, X., Wei, J., Schuurmans, D., Le, Q. V., Chi, E. H., Narang, S., ... & Zhou, D. (2022, September). Self-Consistency Improves Chain of Thought Reasoning in Language Models. In The Eleventh International Conference on Learning Representations.

**Reproducibility:**

4: Could mostly reproduce the results, but there may be some variation because of sample variance or minor variations in their interpretation of the protocol or method.

**Reviewer Confidence:**

4: Quite sure. I tried to check the important points carefully. It's unlikely, though conceivable, that I missed something that should affect my ratings.

---

> ### Author Rebuttal · Authors · 2023-08-29
>
> Thank you for the constructive feedback! To alleviate your concerns raises in “Reasons To Reject”:
>
> > The proposed method requires the user to prompt the LLM to generate pseudo-inputs and answers to each of the test questions, and the user is then required to concatenate the pseudo-demos to the test queries. In contrast, competing methods like AutoCoT or [1] generate a set of pseudo-demos per task but requires the user to provide an unlabelled dataset (which is cheap). On balance, it seems there is still a significant burden on the users, and since the users now need to prompt for model-generated pseudo-inputs and the final predictions, the users need to query the model many more times, so I'm not sure to what extent the claim of "bridging user's practical need" is satisfied.
>
> - The claim “bridging user's practical need” is in the sense that prompting LLMs requires much lower technical barrier for end-users in contrast to execute methods like AutoCoT or [1], which requires collecting unlabelled datasets which may or may not be cheap, usually require a large number of instances, and likely need to be stored locally for their own tasks.
>
> - In fact, Self-ICL can be implemented completely behind the scenes by developers such that users have zero burden and can simply prompt the LLMs as usual by giving only instructional test queries like in most LLM web interfaces. Specifically, suppose a test query is entered by users, developers can easily generate pseudo-inputs, pseudo-labels, and concatenate the pseudo-shots (without exposing to end-users) to the test query via a chain of LLM API calling. Therefore, the users do not need to change their behaviors at all since the nature of Self-ICL is a prompting pipeline, that is, everything can be off-loaded to API chaining by developers.
>
> > Moreover, AutoCoT paper also has a section on the streaming setup where the unlabeled queries arrive in an online fashion; this is a practical enough setup, as we can always collect unlabeled user queries when the model is serving requests. I am requesting the authors to clarify better in what scenarios are their purely zero-shot setup particularly advantageous compared to these previous works?
>
> - Though collecting unlabelled user queries online is a practical enough setup, one of the biggest challenges is that these methods require user’s agreement in advance to obtain and store their queries data. Such preconditions for consent might greatly affect users’ willingness to use the system. Moreover, in scenarios such as medical applications the sensitivity of personal data makes collecting user data infeasible.
> Being purely zero-shot gives Self-ICL advantages over previous works when facing the above-mentioned limitations.
>
> > I wonder whether the authors could 1) state how much more expensive their method is compared to zs-cot and zs-direct, and 2) if the baselines are allowed to use the same amount of compute via self-consistency, is their method still better than the baselines? I think some evidence of this could alleviate the concern that the improvement is simply due to more computation used and can be easily bridged with a simpler alternative.
>
> - Regarding point 1 - For our BBH evaluation, Self-ICL (three shots) is about 7.75 times more expensive than ZS-Direct (zero shot); and 4.12 times more expensive than ZS-CoT (zero shot). The full experimental cost is listed in Appendix A.1.
>
> - Regarding point 2 - Thanks for bringing this up! We have also considered including self-consistency as one of the baselines in our early discussion phase. However, to the best of our knowledge, there were no publicly available results on applying self-consistency to BBH at the time, and due to budget constraints we did not perform such experiments. We agree it would be beneficial to observe the effect of more computation via self-consistency, which we will revisit and try to incorporate in our future version.
>
> > Not a reason to reject, but simply a request for authors to comment: [1] is considered to be a concurrent work, but it considers a very related problem of using ICL in (transductive) zero-shot (similar to AutoCoT). They focus on a different angle in retrieving the best zero-shot demonstrations using a certainty estimate from self-consistency. For example, will the proposed method by the authors benefit from a similar selection procedure? I'd much appreciate if the authors could include discussions of these more recent related works in a revised version of their manuscript.
>
> - Thanks for the suggestion! After reviewing [1], our initial belief is that Self-ICL could further benefit from applying a similar selection procedure like the consistency-based self-adaptive prompting (COSP) proposed in [1]. It is reasonable to assume such a procedure (which is more likely to select “correct” pseudo-labels) can help construct better pseudo-demonstrations and boost Self-ICL’s performance. We are glad to include [1] and other concurrent works as related works and provide a more thorough discussion in our camera-ready paper.

---

### Official Review · Reviewer_EFGJ · 2023-08-01

**Soundness:** 3

**Excitement:**

3: Ambivalent: It has merits (e.g., it reports state-of-the-art results, the idea is nice), but there are key weaknesses (e.g., it describes incremental work), and it can significantly benefit from another round of revision. However, I won't object to accepting it if my co-reviewers champion it.

**Paper Topic And Main Contributions:**

The paper presents a novel approach, called Self-ICL, for zero-shot in-context learning (ICL) using pseudo-inputs as contextual examples. The proposed method involves generating pseudo-inputs and predicting pseudo-labels for these inputs via zero-shot prompting. Using the pseudo-input-label pairs as demonstrations, Self-ICL performs ICL for the test input. The paper demonstrates the superiority of Self-ICL over zero-shot baselines in challenging BIG-Bench Hard tasks. In addition, Self-ICL achieves performance comparable to using real demonstrations by leveraging zero-shot chain-of-thought.

**Reasons To Accept:**

1. Using the generated pseudo-inputs to improve zero-shot ICL performance is rational.
2. The paper is well-written and easy to follow.

**Reasons To Reject:**

1. The idea that using pseudo-inputs as in-context examples is not novel.
2. The paper lacks an in-depth analysis of the impact of various types of pseudo-inputs. For instance, it would be insightful to understand which kind of pseudo-inputs are most effective as in-context examples and how to evaluate their quality.
3. Despite leveraging diversity hints, the language model may generate a mixture of high and low-quality pseudo data (e.g., incorrect pseudo-labels). Therefore, it is crucial to develop methods for automatically filtering out noise data and mitigating the impact of low-quality examples, rather than relying solely on the use of "new," "diverse," and "creative" prompts to enhance diversity.

**Reproducibility:**

4: Could mostly reproduce the results, but there may be some variation because of sample variance or minor variations in their interpretation of the protocol or method.

**Reviewer Confidence:**

3: Pretty sure, but there's a chance I missed something. Although I have a good feel for this area in general, I did not carefully check the paper's details, e.g., the math, experimental design, or novelty.

---

> ### Author Rebuttal · Authors · 2023-08-29
>
> Thank you for the constructive comments! To alleviate your concerns raises in “Reasons To Reject”:
>
> > The idea that using pseudo-inputs as in-context examples is not novel.
>
> - Although there exist previous works leveraging pseudo-inputs as in-context learning examples, their methods differ from ours in several ways as compared in Table 1 and elaborated in Section 6 (Lines 474 - 505). For example, [1][2] requires access to external text corpora/training set for constructing pseudo-inputs, and labels need to be given beforehand in [3]’s approach, thus restricting their applicabilities. On the contrary, Self-ICL does not have any above-mentioned restrictions. Also, a key idea we want to demonstrate is that via self-generated demonstrations, the LLM can leverage its intrinsic capabilities to bootstrap itself for better performance.
>
> - Reference
>    - [1] Lyu, X., Min, S., Beltagy, I., Zettlemoyer, L., & Hajishirzi, H. (2023, February). Z-ICL: Zero-Shot In-Context Learning with Pseudo-Demonstrations. In ICLR 2023 Workshop on Mathematical and Empirical Understanding of Foundation Models.
>    - [2] Zhang, Z., Zhang, A., Li, M., & Smola, A. (2022, September). Automatic Chain of Thought Prompting in Large Language Models. In The Eleventh International Conference on Learning Representations.
>    - [3] Kim, H. J., Cho, H., Kim, J., Kim, T., Yoo, K. M., & Lee, S. G. (2022). Self-generated in-context learning: Leveraging auto-regressive language models as a demonstration generator. arXiv preprint arXiv:2206.08082.
>
> > The paper lacks an in-depth analysis of the impact of various types of pseudo-inputs. For instance, it would be insightful to understand which kind of pseudo-inputs are most effective as in-context examples and how to evaluate their quality.
>
> - Although the current scope of our work focus on demonstrating the LLM’s potential to bootstrap itself by its inner capability, and we have provided analysis along several axes of Self-ICL’s behaviors in Section 5, we agree further investigations focusing on the pseudo-inputs would provide more actionable insights. We will base our analysis in future works on existing literature[1][2][3] discussing how to select representative in-context examples.
>
> - Reference
>    - [1] Sorensen, T., Robinson, J., Rytting, C., Shaw, A., Rogers, K., Delorey, A., ... & Wingate, D. (2022, May). An Information-theoretic Approach to Prompt Engineering Without Ground Truth Labels. In Proceedings of the 60th Annual Meeting of the Association for Computational Linguistics (Volume 1: Long Papers) (pp. 819-862).
>    - [2] Zhang, Y., Feng, S., & Tan, C. (2022, December). Active Example Selection for In-Context Learning. In Proceedings of the 2022 Conference on Empirical Methods in Natural Language Processing (pp. 9134-9148).
>    - [3] Gonen, H., Iyer, S., Blevins, T., Smith, N. A., & Zettlemoyer, L. (2022). Demystifying prompts in language models via perplexity estimation. arXiv preprint arXiv:2212.04037.
>
> > Despite leveraging diversity hints, the language model may generate a mixture of high and low-quality pseudo data (e.g., incorrect pseudo-labels). Therefore, it is crucial to develop methods for automatically filtering out noise data and mitigating the impact of low-quality examples, rather than relying solely on the use of "new," "diverse," and "creative" prompts to enhance diversity.
>
> - Thanks for the instructive feedback, we acknowledge that developing automatic mechanisms capable of selecting ideal pseudo-demonstrations is of great importance to improve Self-ICL’s robustness on prompting formats. We will incorporate this in our future works.

---

### Official Review · Reviewer_B1V1 · 2023-08-07

**Soundness:** 3

**Excitement:**

4: Strong: This paper deepens the understanding of some phenomenon or lowers the barriers to an existing research direction.

**Paper Topic And Main Contributions:**

Large language models (LLMs) have exhibited striking in-context learning (ICL) ability to adapt to target tasks with a few input-output demonstrations. For better ICL, different methods are proposed to select representative demonstrations from existing training corpora. However, such settings are not aligned with real-world practices, as end-users usually query LMs without access to demonstration pools.

This paper introduces Self-ICL---a simple framework which bootstraps LMs' intrinsic capabilities to perform zero-shot ICL. The paper presents the first attempt for true zero-shot ICL that does not require any external data from real distribution or pre-defined label sets.

**Questions For The Authors:**

(1) Is the observation generalizable? Can the proposed idea work on some other models such as LLaMA, Falcon, Flan-T5 or other models in

https://github.com/FranxYao/chain-of-thought-hub

(2) How broadly can the proposed approach work (in which tasks)? When the LLM generate incorrect few shot examples, it can be risky to use the proposed approach, and further use them in the in-context learning. It will be great to include some related discussions of the limitations.

**Reasons To Accept:**

(1) The paper studies an interesting and emerging topic.

(2) The developed solution is simple and works very well.

(3) The developed solution achieves the state-of-the-art performance in some tasks, in the zero-shot setting.

**Reasons To Reject:**

(1) It is unclear how generalizable the observation is. The paper only uses InstructGPT. The paper may also evaluate the proposed idea on some other models such as LLaMA, Falcon, Flan-T5 or other models in

https://github.com/FranxYao/chain-of-thought-hub

The paper may further evaluate the performance in some other common NLP datasets used in tasks such as summarization, text generation, and natural language understanding.

(2) Improvements by the proposed solution are not very surprising in Table 2, in the sense that adding some examples in the in-context learning can outperform the zero-shot prompting. The interesting part might be the part that, the LLM can augment itself by generating some examples.

(3) The paper may also include the results by previous approaches (mentioned in Table 1) in Table 2 for benchmarking purposes (so we understand whether the performances are comparable, or there are big gaps). Sometimes the assumptions made by the previous works in Table 2 are not that restrictive or impractical.

**Reproducibility:**

4: Could mostly reproduce the results, but there may be some variation because of sample variance or minor variations in their interpretation of the protocol or method.

**Reviewer Confidence:**

3: Pretty sure, but there's a chance I missed something. Although I have a good feel for this area in general, I did not carefully check the paper's details, e.g., the math, experimental design, or novelty.

---

> ### Author Rebuttal · Authors · 2023-08-29
>
> Thank you for the constructive feedback! Regarding the points mentioned in “Reasons To Reject”:
>
> > It is unclear how generalizable the observation is. The paper only uses InstructGPT. The paper may also evaluate the proposed idea on some other models such as LLaMA, Falcon, Flan-T5 or other models in ```github.com/FranxYao/chain-of-thought-hub```.
>
> - Thank you for the suggestion! We agree that generalization across different models is important and other instruction-tuned LLMs have been examined in our preliminary experiments, in which promising qualitative results are exhibited. These results are relatively incomplete (in terms of the sample size and number of tasks), and we plan to provide a more comprehensive and systematic evaluation in the future. We adopt InstructGPT as it is one of the most popular and capable LLMs with relatively stable controllability available (during the time we conduct our experiments).
>
> > The paper may further evaluate the performance in some other common NLP datasets used in tasks such as summarization, text generation, and natural language understanding.
>
> - Please refer to the response corresponding to “How broadly can the proposed approach work (in which tasks)?” in "Questions For The Authors".
>
> > Improvements by the proposed solution are not very surprising in Table 2, in the sense that adding some examples in the in-context learning can outperform the zero-shot prompting. The interesting part might be the part that, the LLM can augment itself by generating some examples.
>
> - Indeed, one of the core ideas we want to demonstrate in this work is that via self-generated demonstrations, we can leverage the intrinsic capabilities of LLMs to bootstrap itself for better zero-shot performance. We will emphasize more on this point in our final version.
>
> > The paper may also include the results by previous approaches (mentioned in Table 1) in Table 2 for benchmarking purposes (so we understand whether the performances are comparable, or there are big gaps). Sometimes the assumptions made by the previous works in Table 2 are not that restrictive or impractical.
>
> - Thanks for the suggestion, we will re-examine the restrictiveness and practicality of previous approaches (mentioned in Table 1) for a more comprehensive and contributive evaluation.
>
> Regarding the “Questions For The Authors”:
>
> > Is the observation generalizable? Can the proposed idea work on some other models such as LLaMA, Falcon, Flan-T5 or other models in ```github.com/FranxYao/chain-of-thought-hub```
>
> - Please refer to our responses corresponding to the first point in “Reasons To Reject”.
>
> > How broadly can the proposed approach work (in which tasks)?
>
> - The effectiveness and generalizability of our proposed Self-ICL could be supported based on the coverage of the BIG-Bench Hard (BBH) benchmark used in our evaluation. BBH encompasses a variety of tasks considered hard and beyond common NLU tasks for LLMs to solve, such as data understanding, object tracking, judging geometric shapes, etc. We agree further evaluation and an in-depth analysis comparing performance categorized by the types of tasks is beneficial, and is left as our future work.
>
> > When the LLM generate incorrect few shot examples, it can be risky to use the proposed approach, and further use them in the in-context learning. It will be great to include some related discussions of the limitations.
>
> - For the purpose of boosting end-task performances, as shown in Figure 5, the LLM could still benefit from potentially incorrect examples and outperform the original 0-shot setting (i.e., ZS-Direct). Also, the entire Self-ICL framework can be executed “behind the scenes” so that end-users will not be exposed to incorrect examples. Nevertheless, we agree it is necessary to include more discussions in the limitations section regarding potential risks. We will incorporate this in our final version. Thank you for pointing this out!

---

### Meta-Review · Area_Chair_gEup · 2023-09-19

**Recommendation:** 5

**Metareview:**

Large language models (LLMs) have exhibited striking in-context learning (ICL) ability to adapt to target tasks with a few input-output demonstrations. For better ICL, different methods are proposed to select representative demonstrations from existing training corpora. However, such settings are not aligned with real-world practices, as end-users usually query LMs without access to demonstration pools.
This paper introduces Self-ICL---a simple framework which bootstraps LMs' intrinsic capabilities to perform zero-shot ICL. The paper presents the first attempt for true zero-shot ICL that does not require any external data from real distribution or pre-defined label sets.

Pros:

Well motivated problem, relates to an interesting and relevant topic

Simple intuitive solution of Using the generated pseudo-inputs to improve zero-shot ICL performance

The paper presents the first attempt for true zero-shot ICL that does not require any external data from real distribution or pre-defined label sets.

Cons:

Generalizability of their solution. The paper only uses InstructGPT. The paper may also evaluate the proposed idea on some other models such as LLaMA, Falcon, Flan-T5

Needs to also compare with the methods in Table 1 (AutoCOT, Z_ICL, SG_ICL)  when benchmarking in Table 2

More analysis/ablations on the robustness of the method - e.g. if LLM generates incorrect few shot examples etc

---

### Decision · Program_Chairs · 2023-10-07

**Decision:**

Accept-Main

**Comment:**

Large language models (LLMs) have exhibited striking in-context learning (ICL) ability to adapt to target tasks with a few input-output demonstrations. For better ICL, different methods are proposed to select representative demonstrations from existing training corpora. However, such settings are not aligned with real-world practices, as end-users usually query LMs without access to demonstration pools.
This paper introduces Self-ICL---a simple framework which bootstraps LMs' intrinsic capabilities to perform zero-shot ICL. The paper presents the first attempt for true zero-shot ICL that does not require any external data from real distribution or pre-defined label sets.

Pros:

Well motivated problem, relates to an interesting and relevant topic

Simple intuitive solution of Using the generated pseudo-inputs to improve zero-shot ICL performance

The paper presents the first attempt for true zero-shot ICL that does not require any external data from real distribution or pre-defined label sets.

Cons:

Generalizability of their solution. The paper only uses InstructGPT. The paper may also evaluate the proposed idea on some other models such as LLaMA, Falcon, Flan-T5

Needs to also compare with the methods in Table 1 (AutoCOT, Z_ICL, SG_ICL)  when benchmarking in Table 2

More analysis/ablations on the robustness of the method - e.g. if LLM generates incorrect few shot examples etc